# ICNet: Intra-saliency Correlation Network for Co-Saliency Detection

**Wen-Da Jin**[1*]   **Jun Xu**[2*]   **Ming-Ming Cheng**[2]   **Yi Zhang**[1†]   **Wei Guo**[1]

[1]College of Intelligence and Computing, Tianjin University, Tianjin, China
[2]TKLNDST, CS, Nankai University, Tianjin, China
`{jwd331,yizhang}@tju.edu.cn`, `{csjunxu,cmm}@nankai.edu.cn`

## Abstract

Intra-saliency and inter-saliency cues[3] have been extensively studied for co-saliency detection (Co-SOD). Model-based methods produce coarse Co-SOD results due to hand-crafted intra- and inter-saliency features. Current data-driven models exploit inter-saliency cues, but undervalue the potential power of intra-saliency cues. In this paper, we propose an Intra-saliency Correlation Network (ICNet) to extract intra-saliency cues from the single image saliency maps (SISMs) predicted by any off-the-shelf SOD method, and obtain inter-saliency cues by correlation techniques. Specifically, we adopt normalized masked average pooling (NMAP) to extract latent intra-saliency categories from the SISMs and semantic features as intra cues. Then we employ a correlation fusion module (CFM) to obtain inter cues by exploiting correlations between the intra cues and single-image features. To improve Co-SOD performance, we propose a category-independent rearranged self-correlation feature (RSCF) strategy. Experiments on three benchmarks show that our ICNet outperforms previous state-of-the-art methods on Co-SOD. Ablation studies validate the effectiveness of our contributions. The PyTorch code is available at `https://github.com/blanclist/ICNet`.

## 1  Introduction

Co-Saliency Object Detection (Co-SOD) aims to discover the commonly salient objects in a group of relevant images [36]. It serves as a preliminary step for various computer vision tasks, *e.g.*, co-segmentation [9], co-localization [27], and image retrieval [21], *etc.* The saliency information within a single image (intra-saliency cue) and the occurrence of saliency within a group of images (inter-saliency cue) are essential to the success of existing Co-SOD methods, which can be roughly divided into model-based (non-deep) methods [4, 14, 15] and data-driven (deep) ones [13, 29, 32].

The model-based (non-deep) methods [4, 14, 15] utilize hand-crafted features with manually designed detection pipelines. Most of them leverage as intra cues the single image saliency maps (SISMs) predicted by off-the-shelf SOD methods [5, 44], and compute various inter cues based on subjective priors and hand-crafted features of salient regions in SISMs. Unfortunately, hand-crafted features are usually inconsistent in expressing high-level semantics [36, 43], *e.g.*, versatile viewpoints, complex shapes, and illuminant changes, *etc*, leading to undesirable Co-SOD predictions [8, 14]. Besides, with subjective priors [4, 8], *e.g.*, low-rank constraint, central bias rule, co-saliency distribution consistency and histogram-based contrast, these Co-SOD methods [4, 8] are usually unstable in capturing robust inter cues in complex real-world scenarios [36, 43].

Recently, data-driven (deep) methods [11, 13, 39] are proposed to learn discriminative features with great performance gains on Co-SOD. Among these methods, [38, 39, 40] learn intra and inter cues from scratch to discover similar foregrounds within an image group while distinguishing the foreground and background in each image. The methods of [13, 32] focus on capturing inter cues by imposing various architectures, *e.g.*, recurrent module and group-level concatenation. The works of [10, 11] obliquely take the SISMs as supervisions of the networks and constrain the inter consistency via energy minimization. Despite with promising results, previous Co-SOD networks [28, 29] undervalue the potential power of intra cues for Co-SOD. Recently, the SISMs produced by some SOD networks [35, 42] achieve comparable results with popular Co-SOD networks on Co-SOD benchmarks [2, 33, 37] by standard metrics [1, 3, 6], as will be shown in §4. This indicates that a stronger Co-SOD network can be developed if we well exploit the intra cues in SISMs.

In this paper, we propose an Intra-saliency Correlation Network (ICNet) for fine-grained Co-SOD performance. Our ICNet directly integrates the intra cues and correlation techniques into a deep network for end-to-end learning. We extract intra cues by adopting normalized masked average pooling (NMAP) [25] to combine single image saliency maps (SISMs) predicted by any SOD method and deep features. To explore the inter cues for Co-SOD, we employ a correlation

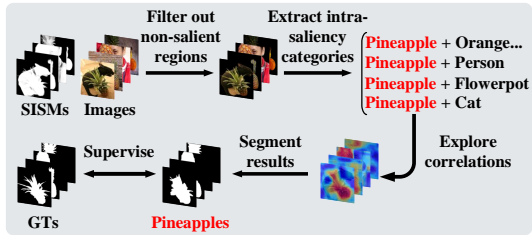

Figure 1: **The main idea of our ICNet.**

fusion module (CFM) to capture the correlations between the extracted intra cues and single-image features. In order to further improve our ICNet on Co-SOD, we design a rearranged self-correlation feature (RSCF) strategy to maintain the feature independence upon semantic categories, while benefiting from global receptive fields. Our main idea is illustrated in Figure 1, from which we can see that co-saliency information can be obtained by exploring correlations between intra-saliency categories. Extensive experiments on three Co-SOD benchmarks demonstrate that our ICNet outperforms state-of-the-art Co-SOD methods on standard objective metrics and subjective visual quality. Ablation studies also validate the effectiveness of each component in our ICNet.

In summary, our major contributions are manifold:

- We propose a novel Intra-saliency Correlation Network (ICNet) for Co-SOD, by integrating intra-saliency features of SISMs and correlation techniques. Ablation studies show that SISMs clearly improve the performance of the proposed ICNet for Co-SOD.

- We validate that well exploiting SISMs improves Co-SOD performance. By leveraging normalized masked average pooling (NMAP) and a correlation fusion module (CFM), intra and inter cues can be well captured from SISM and deep features for Co-SOD.

- We introduce a rearranged self-correlation feature (RSCF) strategy to obtain robust co-saliency features with the inter cues. Benefiting from the independence upon semantic categories and positions, our ICNet with RSCF achieves better Co-SOD performance.

- Experimental results demonstrate that the proposed ICNet outperforms previous state-of-the-art Co-SOD methods on three benchmarks.

## 2   Related Work

Previous model-based Co-SOD methods [4, 14, 15] mainly utilized single image saliency maps (SISMs) produced by off-the-shelf SOD methods as intra-saliency cues, and explored various inter-saliency cues for Co-SOD. The work of [14] measured the similarities between different regions as inter cues, and linearly integrated them with intra cues to derive the co-saliency maps. The method of [15] employed manifold ranking to explore inter cues based on intra cues. Specifically, each image in a group along with its intra cue was utilized to compute correlations with all images in that group. Based on the correlations produced by each pair of images, the inter consistency is extracted to generate final Co-SOD results. Under a low-rank constraint, the method of [4] fused SISMs yielded by multiple SOD models with adaptive weights for Co-SOD predictions. The weights indicate the importance of each SOD model, acting as inter cues to guide the fusion process. However, model-

based Co-SOD methods [4, 14, 15] are limited by hand-crafted features and manually-designed inter cues, which are not robust to complex real-world scenarios.

To alleviate the drawbacks of model-based methods, data-driven Co-SOD methods [13, 29, 32, 39] were proposed to tackle the Co-SOD task with obvious performance gains over previous model-based ones [4, 14, 15]. The work of [32] fed the concatenated features of multiple images into a series of convolutional layers to build a group-level representation, which was further combined with the single-image features for collaborative learning. Later, the authors of [39] proposed an unsupervised learning scheme to derive initial co-saliency masks, which were served as the guidance to train a fully convolutional network [20] for Co-SOD predictions. The method of [29] learned an additional semantic vector in a supervised manner to represent the co-salient category of an image group, boosting the low-level features from the high-level ones for better Co-SOD performance. In [13], the authors sequentially fed single-image features into a recurrent module to progressively update the inter-saliency features, encoding the inter consistency into a robust group-level representation.

Though with remarkable performance gains, some top-tier SOD methods [35, 42] surprisingly achieve comparable results with deep Co-SOD networks [11, 39] on famous benchmarks, as mentioned in [7]. This indicates that if the intra cues in SISMs are well used, we can design a more powerful Co-SOD network. To this end, in this paper, we propose an Intra-saliency Correlation Network to integrate intra-saliency features of SISMs and correlation techniques for fine-grained Co-SOD performance.

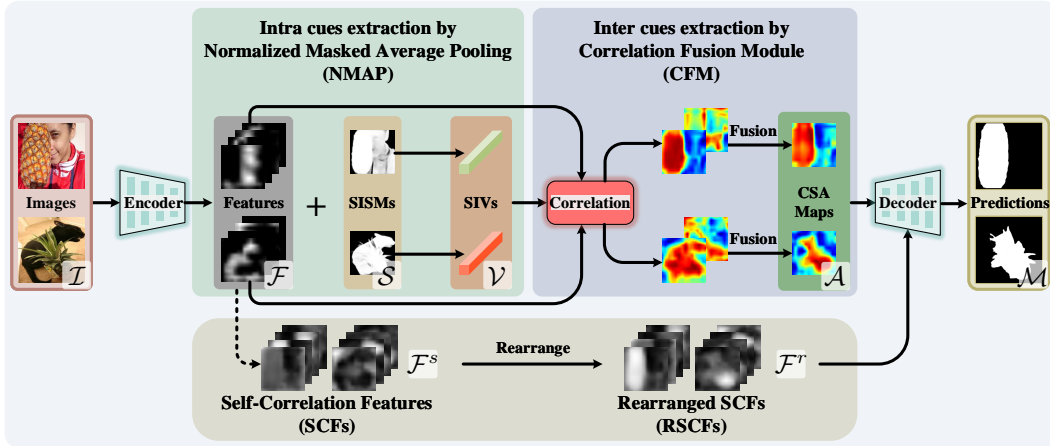

Figure 2: **Pipeline of the proposed ICNet.** We assume the input image group consists of two images to simplify the illustration. In practice, the size of an image group is not fixed. We first utilize NMAP (denoted as "+") to extract intra cues (*i.e.*, SIVs) from the features and corresponding SISMs produced by any off-the-shelf SOD method. Then we employ the CFM to further explore inter cues (*i.e.*, CSA maps) from the intra cues and features by correlation techniques. Finally, these inter cues and the devised RSCFs are integrated to generate co-saliency predictions.

## 3 Proposed Intra-saliency Correlation Network (ICNet)

### 3.1 Overall Network Architecture

Given a group of $n$ relevant images $\mathcal{I} = \{\boldsymbol{I}_i\}_{i=1}^n$, co-saliency object detection (Co-SOD) aims to discover their commonly salient object(s) and generate the co-saliency maps $\mathcal{M} = \{\boldsymbol{M}_i\}_{i=1}^n$. The image group $\mathcal{I}$ is first fed into an encoder network to extract $\ell_2$-normalized high-level semantic features $\mathcal{F} = \{\boldsymbol{F}_i\}_{i=1}^n$. We integrate single image saliency maps (SISMs), denoted as $\mathcal{S} = \{\boldsymbol{S}_i\}_{i=1}^n$, predicted by any SOD method, into a standard deep network for Co-SOD. To explore useful intra cues for Co-SOD, we combine the semantic features $\mathcal{F}$ and corresponding SISMs $\mathcal{S}$, and adopt normalized masked average pooling (NMAP) [25] to produce single-image vectors (SIVs) $\mathcal{V} = \{\boldsymbol{v}_i\}_{i=1}^n$, which represent latent intra-saliency categories (§3.2). To obtain useful inter cues from the intra ones (*i.e.*, SIVs $\mathcal{V}$), we further employ a correlation fusion module (CFM) to exploit correlations between semantic features $\mathcal{F}$ and SIVs $\mathcal{V}$, generating co-salient attention (CSA) maps $\mathcal{A} = \{\boldsymbol{A}_i\}_{i=1}^n$ (§3.3). In order to maintain the consistency of features $\mathcal{F}$ and CSA maps $\mathcal{A}$ in terms of category independence, we propose to compute the self-correlation within features $\mathcal{F}$ and an additional rearranging operation,

obtaining the rearranged self-correlation features (RSCFs) $\mathcal{F}^r = \{\boldsymbol{F}_i^r\}_{i=1}^n$ (§3.4). Finally, the CSA maps and RSCFs are fed into a decoder network to predict the co-saliency maps $\mathcal{M}$ (§3.5). Figure 2 illustrates the pipeline of our ICNet.

## 3.2 Intra Cues Extraction by Normalized Masked Average Pooling

Several deep networks [10, 11] attempt to extract intra cues by taking SISMs as the training targets of a sub-network, rather than directly integrating SISMs into the network for end-to-end training. However, SISMs are not precise enough to indicate the single-salient regions. Thus, explicitly supervising the network training with SISMs would lead to inaccurate intra cues. To better integrate both SISMs and semantic features for more discriminative intra cues, we adopt the normalized masked average pooling (NMAP) operation introduced in [25]. As shown in Figure 3(a), given a group of $\ell_2$-normalized image feature $\mathcal{F} = \{\boldsymbol{F}_i\}_{i=1}^n$ ($\boldsymbol{F}_i \in \mathbb{R}^{C \times H \times W}$), we adjust the corresponding SISMs $\mathcal{S} = \{\boldsymbol{S}_i\}_{i=1}^n$ to proper scales and generate single-image vectors (SIVs) $\mathcal{V} = \{\boldsymbol{v}_i\}_{i=1}^n$ by:

$$\hat{\boldsymbol{v}}_i = \frac{1}{HW} \sum_{x=1}^{H} \sum_{y=1}^{W} \boldsymbol{F}_i(:, x, y) \odot \boldsymbol{S}_i(:, x, y), \boldsymbol{v}_i = \frac{\hat{\boldsymbol{v}}_i}{\|\hat{\boldsymbol{v}}_i\|_2}, \tag{1}$$

where $\odot$ denotes the element-wise multiplication, $x$ and $y$ are the indices along the spatial dimensions. $\|\cdot\|_2$ is the $\ell_2$ norm. Note that we use the SISMs predicted by any off-the-shelf SOD model to directly filter out the features of potentially non-salient regions by multiplication, rather than taking these SISMs as the training targets and forcing the Co-SOD models into overfitting inaccurate SISMs with performance drop. In this way, even though the SISMs are not precisely accurate, the inaccuracy will be largely diluted after averaging and normalizing operations. Thus, $\boldsymbol{v}_i \in \mathbb{R}^C$ is able to express latent intra-saliency categories (Figure 3(b)), and be safely taken as an intra cue.

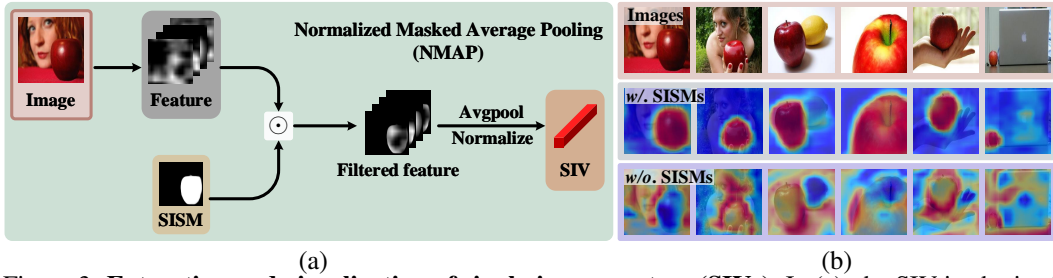

(a)               (b)

Figure 3: **Extraction and visualization of single-image vectors (SIVs)**. In (a), the SIV is obtained by NMAP with the image feature and corresponding SISM. "$\odot$" is the element-wise multiplication. To visualize the semantic feature in the SIV, we compute pixel-level inner products between the SIV and different image features (2-nd row in (b)). The highlighted regions indicate that the SIV could well express the intra-saliency category ("apple") via its semantic feature. However, *w/o.* SISMs, the semantic feature of the produced SIV is meaningless for Co-SOD (3-rd row in (b)).

## 3.3 Inter Cues Extraction by Correlation Fusion Module

To extract inter cues from the intra ones, a naive way is to concatenate SIVs $\mathcal{V}$ with single-image features $\mathcal{F}$, within the framework of existing deep models [13, 29, 32]. However, [32] can only handle an image group with a fixed number of images, [13] is easily influenced by the order of input images due to the recurrent architecture for extracting inter cues, while [29] fails on unseen object categories since the semantic vector is learned on pre-defined categories. To obtain inter cues while avoiding these limitations, we introduce a Correlation Fusion Module (CFM) into our ICNet.

Inspired by video object segmentation community [30], which mainly computes the dense correlations between the features of consecutive frames to achieve fine-grained segmentation. Here, the key to obtaining accurate co-saliency maps is the dense correlations between the SIVs and single-image features. To this end, our CFM computes pixel-level correlations between SIVs in $\mathcal{V}$ and single-image features in $\mathcal{F}$ to generate useful inter cues in parallel, enabling our network to process image groups with any number of images.

To illustrate how our CFM works for the Co-SOD task, we take the $k$-th image $\boldsymbol{I}_k$ from a group of $n$ images as an example. Here we set $n = 4$ for simplicity and better illustration. As demonstrated

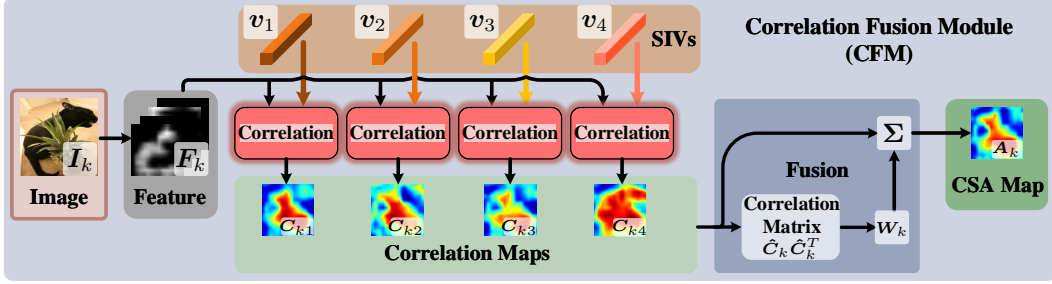

Figure 4: **Correlation fusion module (CFM)**. "$\sum$" denotes weighted summation. For simplicity, here we only calculate the co-salient attention (CSA) map for one image from a group of 4 images. In practice, we implement this process for the feature of each image in the image group to generate multiple CSA maps as inter cues.

in Figure 4, our CFM takes SIVs in $\mathcal{V}$ and the feature $\boldsymbol{F}_k$ of the $k$-th image $\boldsymbol{I}_k$ as the support vectors and the query feature map, respectively. For each SIV $\boldsymbol{v}_i$ in $\mathcal{V}$, we compute the inner product between it and pixel-wise feature vectors in $\boldsymbol{F}_k$, to generate one correlation map $\boldsymbol{C}_{ki} \in \mathbb{R}^{H \times W}$. Each $\boldsymbol{C}_{ki}$ highlights the region of $\boldsymbol{F}_k$ that has high response to the intra-saliency category represented by SIV $\boldsymbol{v}_i$. Nevertheless, for the SIVs that do not represent the co-salient category, the generated correlation maps highlight regions that are irrelevant to the co-salient category. To alleviate the influence of these noisy correlation maps to the final inter cues, we fuse $\{\boldsymbol{C}_{ki}\}_{i=1}^{n}$ with a weight vector that accounts for the relevance between each pair of correlation maps. Specifically, we vectorize and $\ell_2$-normalize each correlation map in $\{\boldsymbol{C}_{ki}\}_{i=1}^{n}$, and then stack them to obtain a matrix $\hat{\boldsymbol{C}}_k \in \mathbb{R}^{n \times HW}$. A weight vector $\boldsymbol{W}_k \in \mathbb{R}^n$ that measures the importance of each correlation map is calculated as follows:

$$\boldsymbol{W}_k = softmax(\alpha \hat{\boldsymbol{C}}_k \hat{\boldsymbol{C}}_k^T \mathbf{1}), \tag{2}$$

where $\alpha$ is a learned factor to regulate the vector to a proper magnitude for the following softmax normalization ($softmax$), $\hat{\boldsymbol{C}}_k \hat{\boldsymbol{C}}_k^T \in \mathbb{R}^{n \times n}$ is a correlation matrix that measures the relevance between every two correlation maps via inner product, while $\mathbf{1}$ represents an $n$-dimensional vector of all ones. With the weight vector $\boldsymbol{W}_k$, we sum the correlation maps $\{\boldsymbol{C}_{ki}\}_{i=1}^{n}$ followed by a min-max normalization and obtain a co-salient attention (CSA) map $\boldsymbol{A}_k \in \mathbb{R}^{H \times W}$ for the feature map $\boldsymbol{F}_k$ as the inter cue. Note that once a correlation map $\boldsymbol{C}_{kj}$ is noisy, it is not similar to most of the other correlation maps, leading to small weight $\boldsymbol{W}_{kj}$. Thus, the weighted fusion suppresses reasonably the noisy correlation maps, enabling the CSA map $\boldsymbol{A}_k$ to discover the potentially co-salient region in $\boldsymbol{F}_k$.

Figure 5 shows some examples of the generated CSA maps. We observe that, the generated CSA maps (3-rd row) highlight the regions that are similar to the co-salient category, although the used SISMs (2-nd row) are noisy to the co-salient category or do not even include any salient objects. This demonstrates that the inter consistency is well expressed by the CSA maps.

### 3.4 Rearranged Self-Correlation Feature

Once we obtain the CSA map $\boldsymbol{A}_k$, we multiply it with the $\ell_2$-normalized feature $\boldsymbol{F}_k$ to focus on the co-salient region and finally predict the Co-SOD map, as suggested in [39]. However, we observe that in this way our network fails to distinguish pixels with similar but different categories, leading to sub-optimal predictions. This is mainly due to the inconsistency on the category dependence between $\boldsymbol{A}_k$ and $\boldsymbol{F}_k$: $\boldsymbol{A}_k$ is category-independent and just reflects the potentially co-saliency scores, while $\boldsymbol{F}_k$ is category-related and each pixel in it is a vector representing a specific category. Specifically, in our initial experiments, we found that the predictions of our ICNet mainly depend on category-independent $\boldsymbol{A}_k$, but the category information (which could be used to further identify the categories of pixels with similar semantics) in $\boldsymbol{F}_k$ is neglected. To tackle this inconsistency, we propose to explicitly utilize the category information in $\boldsymbol{F}_k$ to calculate similarities between pairs of pixels in $\boldsymbol{F}_k$, and transform $\boldsymbol{F}_k$ into category-independent self-correlation feature (SCF). In addition, we extend SCF to a "Rearranged" version (RSCF), further improving the performance of our ICNet on Co-SOD.

**Self-correlation feature (SCF).** Given the feature map $\boldsymbol{F}_k \in \mathbb{R}^{C \times H \times W}$, we reshape it into the size of $C \times HW$, denoted as $\hat{\boldsymbol{F}}_k$. Then we calculate the self-correlation matrix $\hat{\boldsymbol{F}}_k^T \hat{\boldsymbol{F}}_k \in \mathbb{R}^{HW \times HW}$, and reshape it into the size of $HW \times H \times W$ to obtain the SCF $\boldsymbol{F}_k^s$. For the pixel $(x, y)$, regardless of the

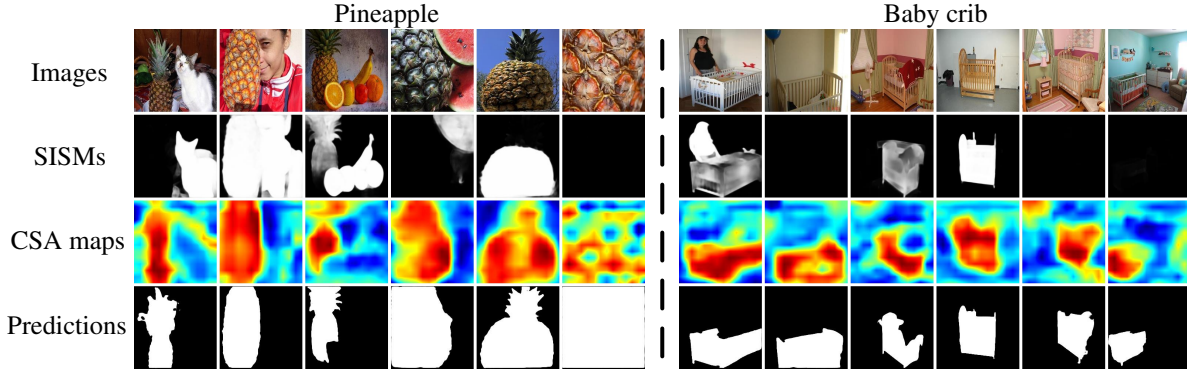

Figure 5: **Visualization of generated co-salient attention (CSA) maps**. The 1-st and 2-nd rows are the input image groups and corresponding SISMs produced by [42], respectively. The 3-rd row shows the CSA maps yielded by our correlation fusion module (CFM). With the CSA maps, our ICNet obtains predictions (4-th row) that are more accurate than the used SISMs.

semantic category expressed by $\boldsymbol{F}_k(:,x,y)$, the counterpart feature vector $\boldsymbol{F}_k^s(:,x,y)$ in SCF only reflects the correlations between $\boldsymbol{F}_k(:,x,y)$ and all pixels of $\boldsymbol{F}_k$, ensuring that the SCF is independent of specific categories. Experiments in §4.3 show that SCF boosts our ICNet on Co-SOD.

**Rearranged SCF (RSCF)**. Though combining SCF with CSA maps benefits from the consistency of category independence, using SCF in our network may potentially lead to the risk of over-fitting. The reason is that each channel of SCF is a self-correlation map related to a certain spatial position, making the learned parameters based on the fixed channel order position-related. To alleviate the over-fitting risk, we rearrange the channel order of SCF. Specifically, for the pixel $(x, y)$ that has higher co-saliency value in $\boldsymbol{A}_k$, the self-correlation map $\boldsymbol{F}_k^s(z,:,:)$ $(z = (x-1)W + y$ is the channel index) will be placed on the upper channel to generate the RSCF $\boldsymbol{F}_k^r$. In this way, the channel order of RSCF is independent of the pixel positions. We will validate the effectiveness of the rearranging operation in §4.3 and provide visual comparisons in the *Supplementary File*.

### 3.5 Implementation Details

We employ the pre-trained VGG-16 [26] as our backbone, and the SISMs are produced by the pre-trained EGNet [42] (also based on VGG-16). To obtain co-saliency maps $\mathcal{M}$, we multiply RSCFs $\mathcal{F}^r$ by CSA maps $\mathcal{A}$ element-wisely to enhance the potentially co-salient regions, generating foreground co-saliency features $\mathcal{F}^f$ with two convolutional layers. Since inter consistency may also exist in the common backgrounds, we perform the above process to obtain background co-saliency features $\mathcal{F}^b$ with the reversion of $\mathcal{S}$. After concatenating foreground features $\mathcal{F}^f$ and background ones $\mathcal{F}^b$, we use two convolution layers to yield the co-saliency enhanced features $\mathcal{F}^c$. In our network, we replace the fully connected layers of VGG-16 [26] with three convolutional layers. For an image group, we extract the $\ell_2$-normalized outputs of 4-th, 5-th and 6-th stages as $\{\mathcal{F}^4, \mathcal{F}^5, \mathcal{F}^6\}$. Then, we obtain corresponding co-saliency enhanced features $\{\mathcal{F}^{c4}, \mathcal{F}^{c5}, \mathcal{F}^{c6}\}$ by performing the above described process on each of $\{\mathcal{F}^4, \mathcal{F}^5, \mathcal{F}^6\}$. Finally, a U-net [24] like decoder is employed to fuse $\{\mathcal{F}^{c4}, \mathcal{F}^{c5}, \mathcal{F}^{c6}\}$ and the low-level features (outputs of 1-st, 2-nd and 3-rd stages of the backbone) to produce final co-saliency maps $\mathcal{M}$.

## 4 Experiments

### 4.1 Experimental Protocol

**Training and test details**. The additional parameters in our proposed modules and the last three layers are initialized with the random normal distribution of which $\mu = 0$, $\sigma = 0.1$. We use Adam [12] as the optimizer to train our ICNet with 60 epochs. The learning rate is $10^{-5}$, and the weight decay is $10^{-4}$. The training set is a subset of the COCO dataset [17], containing 9213 images, as suggested by [13, 32, 43]. All images are resized into $224 \times 224$ in both training and test phases. The training images are randomly flipped horizontally for augmentation. In each training iteration, we randomly select a batch of 10 images from an image group due to limited GPU memory. In the test phase, each image group with an arbitrary number of images constitutes a batch regardless of its

Table 1: **Quantitative comparisons on max F-measure ($F_\beta$), S-measure ($S_\alpha$) and MAE over three benchmark datasets**. "Co" and "Sin" in the "Type" column represent the corresponding methods are Co-SOD models and Single-SOD ones, respectively. "↑" ("↓") means that larger (smaller) is better. The best, second best and third best results are highlighted in <span style="color:red">red</span>, <span style="color:blue">blue</span> and **bold**, respectively.

| Method | Type | Cosal2015 [37] | | | iCoseg [2] | | | MSRC [33] | | |
|---|---|---|---|---|---|---|---|---|---|---|
| | | $F_\beta \uparrow$ | $S_\alpha \uparrow$ | MAE↓ | $F_\beta \uparrow$ | $S_\alpha \uparrow$ | MAE↓ | $F_\beta \uparrow$ | $S_\alpha \uparrow$ | MAE↓ |
| CBCS [8] | Co | 0.539 | 0.546 | 0.234 | 0.728 | 0.673 | 0.165 | 0.639 | 0.516 | 0.279 |
| CSHS [19] | Co | 0.568 | 0.596 | 0.310 | 0.766 | 0.750 | 0.176 | 0.740 | 0.694 | 0.260 |
| CoDW [37] | Co | 0.672 | 0.651 | 0.273 | 0.783 | 0.753 | 0.177 | 0.786 | 0.721 | 0.252 |
| UCSG [10] | Co | 0.745 | 0.756 | 0.158 | **0.836** | 0.825 | 0.117 | 0.847 | **0.806** | 0.164 |
| CSMG [39] | Co | 0.787 | 0.778 | 0.130 | 0.833 | 0.815 | 0.104 | **0.861** | 0.767 | 0.153 |
| MGLCN [11] | Co | **0.791** | 0.804 | 0.128 | <span style="color:blue">0.867</span> | <span style="color:red">0.863</span> | 0.076 | <span style="color:blue">0.862</span> | <span style="color:red">0.816</span> | 0.160 |
| GICD [41] | Co | <span style="color:blue">0.835</span> | <span style="color:blue">0.837</span> | <span style="color:blue">0.072</span> | 0.821 | 0.819 | **0.069** | 0.812 | 0.753 | 0.131 |
| EGNet [42] | Sin | 0.753 | 0.805 | 0.103 | 0.798 | 0.833 | **0.069** | 0.834 | 0.789 | <span style="color:blue">0.122</span> |
| SCRN [35] | Sin | 0.769 | **0.813** | **0.097** | 0.796 | **0.836** | <span style="color:blue">0.066</span> | 0.847 | 0.791 | **0.124** |
| **ICNet** | Co | <span style="color:red">0.860</span> | <span style="color:red">0.855</span> | <span style="color:red">0.058</span> | <span style="color:red">0.874</span> | <span style="color:blue">0.862</span> | <span style="color:red">0.048</span> | <span style="color:red">0.869</span> | <span style="color:blue">0.814</span> | <span style="color:red">0.097</span> |

capacity, and its co-saliency maps are generated at once. The training and test are performed on an Nvidia Titan Xp GPU. Our ICNet is implemented in PyTorch [22] and runs averagely at 80 FPS.

**Loss function**. To well separate the foreground and background, we supervise the predicted co-saliency maps $\mathcal{M} = \{M_i\}_{i=1}^n$ ($M_i \in \mathbb{R}^{H \times W}$) by the corresponding ground-truths $\mathcal{G} = \{G_i\}_{i=1}^n$ ($G_i \in \mathbb{R}^{H \times W}$) under the IoU loss [16]:

$$L(\mathcal{M}, \mathcal{G}) = 1 - \frac{1}{n}\sum_{i=1}^{n} \frac{\sum_{j=1}^{HW}(min(M_i, G_i))_j}{\sum_{j=1}^{HW}(max(M_i, G_i))_j} \tag{3}$$

where $min(\cdot, \cdot)$ and $max(\cdot, \cdot)$ represent the functions that take two maps as inputs and output the element-wise minimum and maximum, respectively. $j$ denotes the pixel position of a map.

**Evaluation metrics**. To quantitatively evaluate the performance of our ICNet, we adopt three widely-used metrics, including max F-measure score [1], S-measure [6] and mean absolute error (MAE) [3].

**Datasets**. We compare our ICNet with state-of-the-art competitors on three popular benchmarks: *MSRC* [33], *iCoseg* [2] and *Cosal2015* [37]. *MSRC* [33] consists of 7 groups of 233 images, and each group has $30 \sim 53$ images with variant co-salient objects. We remove the group "Tree", since several state-of-the-art SOD methods [18, 35, 42] cannot detect any salient object in this group. Note that all the compared methods are evaluated on *MSRC* [33] without the group "Tree" for fairness. *iCoseg* [2] contains 38 groups of 643 images, each group has $4 \sim 41$ images, where the co-salient objects and backgrounds in a group are roughly the same, respectively. *Cosal2015* [37] includes 50 groups of 2015 images, and each group has $25 \sim 52$ images. It is a more challenging benchmark due to the diverse variance in the appearance of co-salient objects with complex backgrounds.

### 4.2 Comparison with State-of-the-arts

**Comparison methods**. We compare our ICNet with seven state-of-the-art Co-SOD methods and two well known Single-SOD ones. For Co-SOD methods, we compare with two model-based (non-deep) CBCS [8] and CSHS [19], and five state-of-the-art data-driven (deep) methods: CoDW [37], UCSG [10], CSMG [39], MGLCN [11], and GICD [41]. We also compare our ICNet with EGNet [42] and SCRN [35], two famous Single-SOD methods.

**Quantitative results** listed in Table 1 show that our ICNet achieves the best results on three Co-SOD benchmarks by three widely used metrics. We note that, 38 out of 45 image groups in *MSRC* and *iCoseg* contain only one category of salient objects, while 43 out of 50 image groups in *Cosal2015* contain multiple categories of salient objects. Thus, the evaluations on *Cosal2015* could reflect better the capability of the comparison methods on Co-SOD than those on *MSRC* and *iCoseg*. Comparing to previous methods, our ICNet obtains large improvements on the challenging *Cosal2015*, but small gains on *MSRC* and *iCoseg*. This demonstrates that our ICNet is very capable of tackling the Co-SOD problem. It also can be seen that, on *MSRC* and *iCoseg*, the gains on $F_\beta$ and $S_\alpha$ are

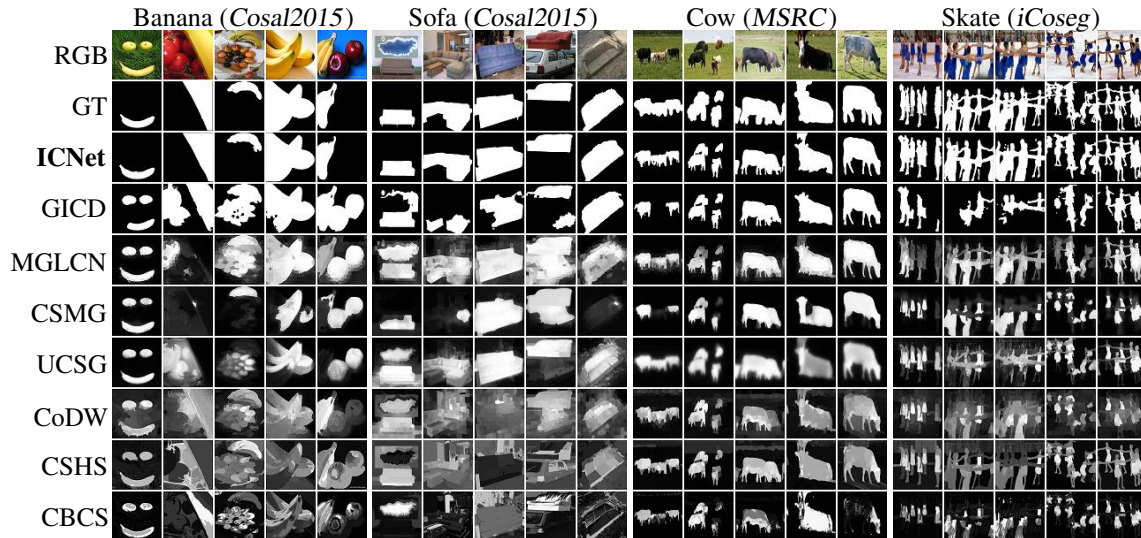

Figure 6: **Qualitative comparisons of different methods** on three benchmark datasets.

more marginal than those on MAE. The reason is that $F_\beta$ is based on the binary saliency map and $S_\alpha$ measures structural similarity, which are not sensitive to the pixel-level incorrectness as MAE.

**Visual comparisons** in Figure 6 demonstrate that, our ICNet utilizes the SISMs well to capture the inter consistency of image groups, and generates more accurate co-saliency maps than other methods.

### 4.3 Ablation Study

Here, we study the effectiveness of each component in our ICNet. All variants have a similar capacity to ensure that the performance gains are not due to the additional parameters.

Table 2: **Results of different variants to our ICNet**. NFs: $\ell_2$-normalized features. CFM: correlation fusion module (§3.3). SISMs: extracting SIVs via SISMs (§3.2). SCFs: self-correlation features. R: rearrange (the channel order of SCFs, §3.4).

| Model | NFs | SCFs | CFM | SISMs | R | Cosal2015 [37] | | | iCoseg [2] | | | MSRC [33] | | |
|---|---|---|---|---|---|---|---|---|---|---|---|---|---|---|
| | | | | | | $F_\beta \uparrow$ | $S_\alpha \uparrow$ | MAE↓ | $F_\beta \uparrow$ | $S_\alpha \uparrow$ | MAE↓ | $F_\beta \uparrow$ | $S_\alpha \uparrow$ | MAE↓ |
| 1 | ✓ | | | | | 0.788 | 0.789 | 0.098 | 0.752 | 0.755 | 0.083 | 0.759 | 0.706 | 0.154 |
| 2 | ✓ | | ✓ | | | 0.815 | 0.823 | 0.081 | 0.726 | 0.763 | 0.081 | 0.782 | 0.742 | 0.139 |
| 3 | ✓ | | ✓ | ✓ | | 0.824 | 0.833 | 0.076 | 0.864 | 0.862 | 0.047 | 0.830 | 0.789 | 0.114 |
| 4 | | ✓ | | | | 0.767 | 0.787 | 0.105 | 0.804 | 0.804 | 0.070 | 0.772 | 0.717 | 0.148 |
| 5 | | ✓ | ✓ | | | 0.817 | 0.824 | 0.078 | 0.813 | 0.815 | 0.067 | 0.839 | 0.789 | 0.113 |
| 6 | | ✓ | ✓ | ✓ | | 0.855 | 0.850 | 0.060 | 0.857 | 0.848 | 0.048 | 0.872 | 0.807 | 0.100 |
| 7 | | ✓ | ✓ | ✓ | ✓ | 0.860 | 0.855 | 0.058 | 0.874 | 0.862 | 0.048 | 0.869 | 0.814 | 0.097 |

**Effectiveness of intra and inter cues**. In our ICNet, we exploit SISMs to represent intra cues by single image vectors (SIVs) as described in §3.2, and capture inter cues by our CFM introduced in §3.3. Here we validate the effectiveness of SISMs and CFM for Co-SOD. For our ICNet without "SISMs", we replace the SISMs with masks of all ones to build SIVs. For our ICNet without "CFM", we average the group of SIVs to a group-level vector in $\mathbb{R}^C$ and copy it for $H \times W$ times to form a tensor in $\mathbb{R}^{C \times H \times W}$ as the inter cue, and concatenate it with each single-image feature to predict co-saliency maps [29, 32]. As shown in Table 2, by comparing the results of models "1" and "2" as well as models "4" and "5", we observe that our ICNet with CFM obtains better performance. This indicates that the CFM captures the inter consistency from the SIVs better than simple concatenation. Equipped with SISMs, the models "3" and "6" outperform models "2" and "5" on all three metrics respectively, since filtering out non-salient regions in semantic features with SISMs makes the generated SIVs express more accurate intra-saliency categories. In summary, with SISMs and CFM, our ICNet achieves fine-grained Co-SOD by exploiting discriminative intra and inter cues.

**Importance of RSCF to our ICNet**. To study this problem, we first replace the $\ell_2$-normalized features ("NFs") with our proposed SCFs. In Table 2, the comparisons between the results of models "1" and "4", "2" and "5", as well as "3" and "6" already show the effectiveness of SCFs. By comparing

the results of models "6" and "7", we observe that the rearranging operation ("R") further boosts the results slightly, since the potential dependence on position is eliminated.

**How much does our ICNet improve basic SOD models**? To study this question, we conduct experiments based on a model-based SOD method GC [5] and 4 data-driven SOD methods: EGNet [42], BASNet [23], CPD [34] and F3Net [31]. As listed in Table 3, our ICNet achieves clearly better results than the baselines, no matter which SOD method is used to generate the SISMs. This indicates the effectiveness and robustness of our ICNet upon the SISMs produced by different SOD methods.

Table 3: **Results of our ICNet with SISMs by various SOD methods**. "Baseline": the basic Single-SOD methods. "ICNet": our ICNet with SISMs produced by corresponding SOD methods.

| Method | Cosal2015 [37] | | | | | | iCoseg [2] | | | | | | MSRC [33] | | | | | |
| | Baseline | | | ICNet | | | Baseline | | | ICNet | | | Baseline | | | ICNet | | |
| | $F_\beta\uparrow$ | $S_\alpha\uparrow$ | MAE$\downarrow$ | $F_\beta\uparrow$ | $S_\alpha\uparrow$ | MAE$\downarrow$ | $F_\beta\uparrow$ | $S_\alpha\uparrow$ | MAE$\downarrow$ | $F_\beta\uparrow$ | $S_\alpha\uparrow$ | MAE$\downarrow$ | $F_\beta\uparrow$ | $S_\alpha\uparrow$ | MAE$\downarrow$ | $F_\beta\uparrow$ | $S_\alpha\uparrow$ | MAE$\downarrow$ |
|---|---|---|---|---|---|---|---|---|---|---|---|---|---|---|---|---|---|---|
| GC [5] | 0.640 | 0.682 | 0.149 | 0.854 | 0.852 | 0.064 | 0.698 | 0.708 | 0.124 | 0.859 | 0.853 | 0.056 | 0.755 | 0.691 | 0.160 | 0.878 | 0.826 | 0.093 |
| EGNet [42] | 0.753 | 0.805 | 0.103 | 0.860 | 0.855 | 0.058 | 0.798 | 0.833 | 0.069 | 0.874 | 0.862 | 0.048 | 0.834 | 0.789 | 0.122 | 0.869 | 0.814 | 0.097 |
| BASNet [23] | 0.790 | 0.821 | 0.096 | 0.858 | 0.849 | 0.059 | 0.855 | 0.866 | 0.054 | 0.862 | 0.852 | 0.053 | 0.869 | 0.821 | 0.098 | 0.867 | 0.810 | 0.101 |
| CPD [34] | 0.761 | 0.807 | 0.100 | 0.857 | 0.851 | 0.060 | 0.802 | 0.834 | 0.063 | 0.857 | 0.850 | 0.051 | 0.849 | 0.791 | 0.116 | 0.870 | 0.815 | 0.097 |
| F3Net [31] | 0.810 | 0.840 | 0.084 | 0.856 | 0.850 | 0.059 | 0.844 | 0.864 | 0.052 | 0.864 | 0.855 | 0.048 | 0.864 | 0.822 | 0.103 | 0.865 | 0.813 | 0.097 |

**Impacts of the batch size to our ICNet**. In the default setting, we train our ICNet with batch size of 10 but take the whole image group as a batch for the test, which leads to the inconsistency between the batch size of "$n_{train}$" and "$n_{test}$" in the training and test stage, respectively. To explore the impacts of this inconsistency and different settings of batch size to the final results, we evaluate the performance of our ICNet under multiple settings in Table 4. When $n_{train}$ is set as 5/10/15, in the test stage we use all images in a group as a batch. When $n_{test}$ is set as 5/10/20, in the training stage we set $n_{train} = 10$. The results show that the training batch size influences little our ICNet and the inconsistent batch sizes in training and test phases are empirically available. Besides, our ICNet leverages SISMs better when taking the whole group as input during tests, as increasing the test batch size brings slight improvements on *Cosal2015* [37].

Table 4: **Results of our ICNet with different batch size settings**. "$n_{train}$" and "$n_{test}$" denote training and test batch size, respectively.

| Model | Cosal2015 [37] | | | iCoseg [2] | | | MSRC [33] | | |
| | $F_\beta\uparrow$ | $S_\alpha\uparrow$ | MAE$\downarrow$ | $F_\beta\uparrow$ | $S_\alpha\uparrow$ | MAE$\downarrow$ | $F_\beta\uparrow$ | $S_\alpha\uparrow$ | MAE$\downarrow$ |
|---|---|---|---|---|---|---|---|---|---|
| $n_{train} = 5$ | 0.858 | 0.854 | 0.058 | 0.866 | 0.854 | 0.050 | 0.868 | 0.812 | 0.098 |
| $n_{train} = 10$ | 0.860 | 0.855 | 0.058 | 0.874 | 0.862 | 0.048 | 0.869 | 0.814 | 0.097 |
| $n_{train} = 15$ | 0.857 | 0.850 | 0.060 | 0.863 | 0.853 | 0.052 | 0.876 | 0.813 | 0.096 |
| $n_{test} = 5$ | 0.844 | 0.846 | 0.066 | 0.865 | 0.856 | 0.050 | 0.870 | 0.815 | 0.097 |
| $n_{test} = 10$ | 0.851 | 0.849 | 0.063 | 0.874 | 0.862 | 0.048 | 0.869 | 0.814 | 0.097 |
| $n_{test} = 20$ | 0.856 | 0.852 | 0.061 | 0.873 | 0.862 | 0.048 | 0.869 | 0.815 | 0.097 |

### 4.4 Failure Case Study

Our ICNet is heavily based on the SISMs, and fails on Co-SOD when the used SISMs are unreliable. For example, the SISMs of the "Tree" category (in *MSRC* [33]) produced by EGNet [42] contain negligible saliency information, making our ICNet produce meaningless predictions on this case. We illustrate this point in the *Supplementary File* due to limited space.

## 5 Conclusion

In this paper, we proposed an Intra-saliency Correlation Network (ICNet) for co-saliency detection (Co-SOD). By directly integrating single image saliency maps produced by any off-the-shelf SOD method into the deep neural network for discriminative intra cues extraction, we further exploited correlations between intra cues and single-image features to capture accurate inter cues for Co-SOD. Besides, by leveraging correlations within the image features, we devised a Rearranged Self-Correlation Feature strategy combined with the inter cues, to further boost our ICNet on Co-SOD. Experiments demonstrated that our ICNet achieves better performance than state-of-the-art Co-SOD methods on three benchmarks. Comprehensive ablation studies also validated our contributions.

## Broader Impact

This work would potentially benefit researchers in the computer vision community, especially those who are interested in (co-)saliency detection. The authors believe that this work does not have negative societal consequences.

## Acknowledgments and Disclosure of Funding

We thank the reviewers for their valuable suggestions and insightful comments. This work was supported in part by the Major Project for New Generation of AI under Grant No. 2018AAA0100400, National Natural Science Foundation of China (61702359, 61922046), and Tianjin Natural Science Foundation (18ZXZNGX00110).

## Footnotes

*Wen-Da Jin and Jun Xu are joint first authors.

†Yi Zhang is the corresponding author.

[3]To simplify writing, we abbreviate intra-saliency and inter-saliency cues as intra and inter cues, respectively.

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
