[Supplementary Material]

# Supplementary File to "ICNet: Intra-saliency Correlation Network for Co-Saliency Detection"

Paper ID 1565

## 1 Content

In this supplementary file, we provide more details of our Intra-saliency Correlation Network (ICNet), as well as more comparisons with other Co-SOD methods to illustrate the superiority of our ICNet. Specifically,

- in §2, we present visualizations of Rearranged Self-Correlation Feature (RSCF);

- in §3, we provide visual results of the failure case;

- in §4, we show more experimental results to compare our ICNet with other methods.

## 2 Visualizations of RSCF

In this section, we provide visualizations of Self-Correlation Features (SCFs) and Rearranged SCFs (RSCFs) to better illustrate how our "Rearrange" operation eliminates the position dependence of SCF.

As mentioned in our main paper, each channel of SCF (*i.e.*, $F_k^s(z, :, :)$) is a self-correlation map measuring the correlation between the feature vector $F_k(:, x, y)$ ($z = (x - 1)W + y$) and all feature vectors in $F_k$, making the self-correlation map $F_k^s(z, :, :)$ related to the specific spatial position $(x, y)$. In Figure 1, we show some self-correlation maps (2-nd row) and their related positions (white spots in 1-st row). Since each channel of SCF and the related position are one-to-one while the channel order of SCF is fixed, the learned parameters on SCF are position-related. In real-world scenarios, the positions of co-salient objects in images are random. However, once the training dataset has position biases on the co-salient objects, the position-related parameters (due to SCF) would be influenced by these biases, leading to the potential risk of overfitting.

To avoid this risk, we rearrange the channel order of SCF to ensure that the parameters are learned without position information. In practice, given the SCF $F_k^s \in \mathbb{R}^{HW \times H \times W}$, we rearrange its channel order according to the potentially co-saliency values in the CSA map $A_k$. Specifically, for the pixel position $(x, y)$, if the value $A_k(x, y)$ (the number in each image in the 1-st row of Figure 1) is higher, the corresponding self-correlation map $F_k^s(z, :, :)$ ($z = (x - 1)W + y$ is the channel index) will be placed on the upper channel to generate the Rearranged SCF (RSCF). As shown in Figure 1, after the rearranging operation, the self-correlation maps in SCFs that are more related to the co-salient category are aggregated in the upper channels of RSCFs, and the rearranged channel orders are not fixed due to the different CSA maps. In this way, our ICNet learns to utilize the co-saliency information encoded in the rearranged channel order of RSCF, rather than the position information with potential biases encoded in the fixed channel order of SCF, for Co-SOD.

## 3 Visualizations of the Failure Case

To further illustrate the failure cases of our ICNet, we show the visual comparisons on the "Tree" category of *MSRC* [8] in Figure 2. We can observe that the SISMs produced by EGNet [12] (3-rd row) contain almost no saliency information. Based on these extremely unreliable SISMs, the SIVs built by NMAP express meaningless co-salient category information. Thus, our ICNet fails to utilize such ambiguous intra cues to further explore inter consistency, leading to the failure predictions (4-th row). On the contrary, MGLCN [5] and UCSG [4] perform better on this image group, because they are not based on SISMs produced by off-the-shelf SOD models. Although this failure case may raise doubts about the robustness of our ICNet, this extreme case rarely happens. Meanwhile, our experiments have shown that ICNet can perform robustly as long as not mostly SISMs are unreliable.

# 4 More Experimental Results

In this section, we provide more experimental results on F-measure [1] (§4.1) and visualizations (§4.2), to further compare our ICNet with other Co-SOD methods: CBCS [3], CSHS [6], CoDW [9], UCSG [4], GW [7], CSMG [10], MGLCN [5] and GICD [11].

## 4.1 More comparisons on F-measure

In our main paper, we adopt the widely used max F-measure [1] as one important metric to evaluate our ICNet. However, max F-measure only shows the quality of the predicted co-saliency maps (in binary values) at a preset threshold, which can not comprehensively demonstrate the capability of our ICNet. To this end, as shown in Figure 3, we plot the curves of F-measures with different thresholds. We observe that, although CSMG [10] and MGLCN [5] have comparable max F-measures with our ICNet on *MSRC* [8] and *iCoseg* [2], once the threshold is changed, the F-measures of these methods drop significantly. On the contrary, no matter which threshold is selected to binarize the co-saliency maps, our ICNet achieves stable while high F-measures on the *MSRC* [8] and *iCoseg* [2], as well as the challenging *Cosal2015* [9] datasets. Since its predictions have consistent values in foregrounds, our ICNet obtains satisfying binary co-saliency maps without a deliberately considered threshold. This demonstrates the superiority of our ICNet over previous Co-SOD methods.

## 4.2 More comparisons on visual results

In Figures 4 - 10, we provide more visual results to qualitatively compare our ICNet with other methods on the three benchmarks. One can see that, our ICNet obtains more accurate and fine-grained co-saliency maps than the other competitors.

Figure 1: **Visualizations of RSCFs with the co-saliency category "pineapple"**. "Original order" and "Rearranged order" denote the channel index of the self-correlation maps (2-nd row of subfigures) in SCFs and RSCFs, respectively. To visualize the position dependence of SCF, we superimpose the CSA map on top of the image (1-st row of subfigures), and use the white spot to indicate the position $(x, y)$ related to the self-correlation map $\boldsymbol{F}_k^s(z, :, :)$ in SCF, where $z = (x - 1)W + y$. In the corners there are the values of these related positions in the corresponding CSA maps, which are utilized to determine the rearranged channel orders.

Figure 2: **Visual comparisons on the "Tree" category of *MSRC* [8]**. Based on the meaningless SISMs produced by EGNet [12] (3-rd row), our ICNet fails to detect co-salient object(s) (4-th row).

(a) *MSRC* [8]

(b) *iCoseg* [2]

(c) *Cosal2015* [9]

Figure 3: **Comparisons on F-measures at different thresholds** of our ICNet and other Co-SOD methods over the three benchmark datasets.

Figure 4: **Visual comparisons** of our ICNet and other Co-SOD methods on *MSRC* [8].

Kite                                    Plane

RGB

GT

**ICNet**

GICD [11]

MGLCN [5]

CSMG [10]

UCSG [4]

CoDW [9]

CSHS [6]

CBCS [3]

Figure 5: **Visual comparisons** of our ICNet and other Co-SOD methods on *iCoseg* [2].

Figure 6: **Visual comparisons** of our ICNet and other Co-SOD methods on *iCoseg* [2].

Figure 7: **Visual comparisons** of our ICNet and other Co-SOD methods on *Cosal2015* [9].

Figure 8: **Visual comparisons** of our ICNet and other Co-SOD methods on *Cosal2015* [9].

Figure 9: **Visual comparisons** of our ICNet and other Co-SOD methods on *Cosal2015* [9].

Figure 10: **Visual comparisons** of our ICNet and other Co-SOD methods on *Cosal2015* [9].