[Reviews · NeurIPS 2020]

Review 1

Summary and Contributions: The co-saliency task is addressed in the paper. The authors consider the intra- and inter-image saliency together to improve the co-saliency results. The former is extracted with normalized masked average pooling and pre-computed single-image saliency maps. For the later, the correlation operator is applied to all images to capture the inter-image cues. Besides, to address the problem of the model that can't recognize objects with similar semantics, the self-feature scheme is proposed the category-independent signals to refine the co-saliency maps. In the experimental results, the proposed method outperforms the current state-of-the-art methods, especially on Cosal2015 dataset. Besides, the detailed ablation studies are conducted to validate the robustness of the proposed method.

Strengths: 1. The proposed method is reasonable. The proposed rearranged self-correlation feature is novel and interesting, and it can contribute to the large performance gain. 2. The paper is clearly presented and well-organized. 3. The proposed method outperforms the current state-of-the-art methods, and the detailed ablation studies are conducted to validate the robustness of the proposed method.

Weaknesses: 1. Some related works missing There are some recent related works, such as [Ref. 1~Ref.3], and it is better to cite these papers and have some discussion. [Ref. 1] Zhang et al., "Adaptive Graph Convolutional Network with Attention Graph Clustering for Co-saliency Detection," CVPR'20 [Ref. 2] Fang et al., "Taking a Deeper Look at Co-Salient Object Detection," CVPR'20 [Ref .3] Tsai et al., "Deep Co-saliency Detection via Stacked Autoencoder-enabled Fusion and Self-trained CNNs," TMM'19 2. Overclaimed contribution The proposed method contains many components, such as the combination of intra- and inter-image saliency, correlation fusion module, the normalized masked average pooling, rearranged self-correlation feature. However, the combination of intra- and inter-image saliency is done in [9], and the correlation fusion module is adopted in [9], too. Besides, the normalized masked average pooling has been proposed in [23]. These three should not be claimed or emphasized as the paper's contribution, but the authors should emphasize the rearranged self-correlation component. The proposed method is similar t to [9], which considers both intra- and inter-image saliency and the correlation module. The major differences are the normalized masked average pooling and rearranged self-correlation features, but the former has been proposed in [23]. Therefore, I think only considering rearranged self-correlation, the novelty is somewhat not enough to be accepted. 3. About the experimental results 3-1. The proposed method got a significant performance gain in Cosal 2015 dataset, but in the other two datasets, the only small performance gain is achieved. The authors should give a detailed discussion about it. 3-2. The model is trained with ten images from an image group, but the co-saliency maps could be generated for an arbitrary number of images. The training and inference seem not consistent. What are the results of the different images in an image group for training?

Correctness: Yes

Clarity: Yes

Relation to Prior Work: No. Please see Weaknesses

Reproducibility: No

Additional Feedback:


Review 2

Summary and Contributions: This work presents a Co-SOD method by utilizing the abundant information from off-the-shelf SOD method for extracting intra cues and deeply exploring inter cues in feature-level among an image group. The contributions mainly focus on the exploring intra and inter cues in a single image groups under the reference of SISM, and enhance this ability by RSCF.

Strengths: This paper skillfully explores intra and inter cues by several modules and achieves appealing performance which exceeds previous works a large margin in MAE metric. The proposed CFM module leverages SISMs as references, extracts intra cues in SIVs and explores inter cues (CSA maps) among an image group. The ablation studies and experimental results prove the effectiveness of the proposed modules.

Weaknesses: 1. The ablation studies base on Cosal2015 dataset and show convincing results. However, this work only obtains large improvement in this dataset while get tiny one in others, especially in terms of F-measure and S-measure. So, will this method also prove itself by performing ablation on other datasets? By the way, I cannot get your idea about the ablation setting for the CFM module, that is, how to concatenate a “group-level” vector with feature? 2. According to figures shown in this work, it’s hard to believe that “inter consistency may also exist in the common background”, especially in Banana, Sofa and Pineapple; Guitar, Hammer, Bowl (in supplementary file), etc. So, whether this operation for background really makes sense is a doubt to me. 3. The group “Tree” is removed because SOD methods behave badly on it, but most other data-driven methods don’t take SISM for reference, if these methods also ignore “Tree”? If not, experiments comparison on this dataset may need a supplementary or something.

Correctness: Somewhat yes.

Clarity: Somewhat yes.

Relation to Prior Work: Somewhat yes

Reproducibility: No

Additional Feedback: Some of my concerns are addressed. This work need to add some ablation study in manucsript. In addition, the author have not clearly explain the question 3. Therefore, i will keep my rating.


Review 3

Summary and Contributions: This paper proposes an Intra-saliency Correlation Network (ICNet) to extract intra-saliency cues from the single image saliency maps (SISMs) and obtain inter-saliency cues by correlation techniques. Specifically, the authors first adopt a normalized masked average pooling (NMAP) technique to extract latent intra-saliency categories from the SISMs and semantic features as intra cues. Then, they design a correlation fusion module (CFM) to obtain inter cues by exploiting correlations between the intra cues and single-image features. Besides, the authors also propose a category-independent rearranged self-correlation feature (RSCF) strategy to further improve the Co-SOD performance.

Strengths: + The strategy of using the correlation to model inter-saliency sounds good and reasonable. It is a natural choice to model inter-saliency patterns across the image group. +Good experimental results are obtained.

Weaknesses: -It seems that the proposed learning system would be influenced by the number of images in each image group. Th authors are suggested to discuss how big the influence would be. -As the SISMs of this method are obtained from the most recent SOD methods, it is not so fair to compare this approach to the existing methods. -It is not clear why Rearranged SCF works. -Although the whole idea is sound, the proposed approach combines many existing techniques, such as off-the-shelf SOD method, NMAP, non-local feature correlation, etc. This hurts the technical novelty of this work.

Correctness: Most claims look correct and the methodology is reasonable.

Clarity: The paper is well written and easy to follow.

Relation to Prior Work: The difference between this work and the previous works is clearly discussed.

Reproducibility: No

Additional Feedback: ----------------------------update after rebuttal--------------------------- Very appreciate for the authors' efforts to provide the response. Some of my concerns are addressed while some are still remind. Thus, I keep my original rating.


Review 4

Summary and Contributions: This paper develops a co-saliency detection method that exploits intra- and inter-saliency cues. It integrates intra-saliency features of single image saliency maps (SISMs) and designs a correlation fusion module (CFM) to exploit their correlations. A rearranged self-correlation feature (RSCF) strategy is proposed to obtain robust co-saliency features from inter-saliency cues. The experiments and ablation studies on three co-saliency benchmarks demonstrate effectiveness of intra- and inter-saliency cues, as well as the proposed modules.

Strengths: This paper is well motivated from the observation that salient object detection methods achieve comparable performances over co-saliency methods. Utilizing SISMs for better intra-saliency cue extraction is a novel contribution. The RSCF strategy can effectively improve the consistency between category-independent co-salient attention maps and category-related image features. Experimental results are adequate and comprehensive.

Weaknesses: This paper adopts normalized masked average pooling (NMAP) from [23] and dense correlations from [28]. Thus, the main technical novelties is the SCF and its rearranged variant. While the author claims that directly integrating SISMs generated by any off-the-shelf SOD model is better than taking SISMs as the training targets, it is still unclear why the former is better since both of them utilize inaccurate SISMs. It is also unclear why adding pooling/normalization layers for the latter cannot dilute the inaccuracy. Though the improvement of MAE is significant, the F-measure and S-measure on MSRC and iCoseg is marginal for ICNet. It is suggested to include an analysis about the possible reasons that degrade the model performance on these metrics.

Correctness: Both the claim and the method is correct.

Clarity: The writing of this paper is good.

Relation to Prior Work: This work clearly discussed how it is different from previous works.

Reproducibility: Yes

Additional Feedback:

[Author Response · NeurIPS 2020]

Table 1

| Model | MSRC | | | iCoseg | | | CoSal2015 | | |
|---|---|---|---|---|---|---|---|---|---|
| | $F_\beta\uparrow$ | $S_\alpha\uparrow$ | MAE↓ | $F_\beta\uparrow$ | $S_\alpha\uparrow$ | MAE↓ | $F_\beta\uparrow$ | $S_\alpha\uparrow$ | MAE↓ |
| $n_{train}=5$ | 0.868 | 0.812 | 0.098 | 0.866 | 0.854 | 0.050 | 0.858 | 0.854 | 0.058 |
| $n_{train}=10$ | 0.869 | 0.814 | 0.097 | 0.874 | 0.862 | 0.048 | 0.860 | 0.855 | 0.058 |
| $n_{train}=15$ | 0.876 | 0.813 | 0.096 | 0.863 | 0.853 | 0.052 | 0.857 | 0.850 | 0.060 |
| $n_{test}=5$ | 0.870 | 0.815 | 0.097 | 0.865 | 0.856 | 0.050 | 0.844 | 0.846 | 0.066 |
| $n_{test}=10$ | 0.869 | 0.814 | 0.097 | 0.874 | 0.862 | 0.048 | 0.851 | 0.849 | 0.063 |
| $n_{test}=20$ | 0.869 | 0.815 | 0.097 | 0.873 | 0.862 | 0.048 | 0.856 | 0.852 | 0.061 |
| ICNet with GC | 0.867 | 0.808 | 0.098 | 0.863 | 0.858 | 0.049 | 0.845 | 0.846 | 0.065 |

Table 2

| Model | NFs | SCFs | SIVs | CFM | R | MSRC | | | iCoseg | | |
|---|---|---|---|---|---|---|---|---|---|---|---|
| | | | | | | $F_\beta\uparrow$ | $S_\alpha\uparrow$ | MAE↓ | $F_\beta\uparrow$ | $S_\alpha\uparrow$ | MAE↓ |
| 1 | ✓ | | | | | 0.698 | 0.673 | 0.164 | 0.643 | 0.678 | 0.131 |
| 2 | ✓ | | ✓ | | | 0.762 | 0.709 | 0.154 | 0.779 | 0.773 | 0.084 |
| 3 | ✓ | | ✓ | ✓ | | 0.840 | 0.778 | 0.112 | 0.845 | 0.846 | 0.061 |
| 4 | | ✓ | | | | 0.837 | 0.785 | 0.115 | 0.771 | 0.786 | 0.079 |
| 5 | | ✓ | ✓ | | | 0.844 | 0.793 | 0.113 | 0.830 | 0.831 | 0.065 |
| 6 | | ✓ | ✓ | ✓ | | 0.855 | 0.796 | 0.108 | 0.856 | 0.844 | 0.056 |
| 7 | | ✓ | ✓ | ✓ | ✓ | 0.869 | 0.814 | 0.097 | 0.874 | 0.862 | 0.048 |

**Novelty (R1/R3/R4)**. The key contribution of our work is that we exploited SISMs in a better manner, and obtained
discriminative intra-saliency cues for better Co-SOD performance. To this end, we directly integrated SISMs (produced
by previous SOD method) as intra-saliency priors (instead of training targets) into the deep neural network to extract
intra cues, and exploited the correlation information in the intra cues to obtain inter cues for detecting co-saliency.
What's more, we proposed the RSCF module to further improve our ICNet on Co-SOD. Experiments demonstrated
that our ICNet well leverages SISMs and becomes a new SOTA method for Co-SOD. Ablation studies also validated
the effectiveness of RSCF in **Lines 269-278** of the main paper. We do not take the NMAP and CFM modules as our
contributions. We have claimed our contributions more clearly in the revised submission, and thanks for the suggestions.

**Performance on *MSRC* and *iCoseg* (R1/R4)**. In *MSRC* and *iCoseg*, 38 out of 45 image groups contain only one
category of salient objects. In *CoSal2015*, 43 out of 50 image groups contain more than one category of salient objects.
Thus, the evaluations on *CoSal2015* could reflect better the capability of the comparison methods on Co-SOD than
those on *MSRC* and *iCoseg*. Comparing to previous methods, our ICNet obtains large improvements on *CoSal2015*, but
small gains on *MSRC* and *iCoseg*. This demonstrates that our ICNet is very capable of tackling the Co-SOD problem.
$F_\beta$ is based on the binary saliency map and $S_\alpha$ measures structural similarity, they are not sensitive to the pixel-level
incorrectness as MAE. Hence, our ICNet outperforms marginally over the other methods on $F_\beta$ and $S_\alpha$.

**How the size of the image group influences our ICNet (R1/R3)**? In Table 1, we summarize the results of our ICNet
with a group size of "$n_{train}$" or "$n_{test}$" in the training or test stage, respectively. When $n_{train}$ is set as 5/10/15, in the
test stage we use all images in a group. When $n_{test}$ is set as 5/10/20, in the training stage we set $n_{train}=10$. The
results show that the training group size influences little our ICNet and the inconsistent group sizes in training and test
phases are empirically available. Our ICNet leverages SISMs better when taking the whole group as input during tests,
as increasing the test group size brings slight improvements on CoSal2015.

**Related works (R1)**. We have introduced the suggested related works in the revision of our submission.

**Ablation studies on *MSRC* and *iCoseg* (R2)**. Table 2 shows that our model is also effective on *MSRC* and *iCoseg*.

**Models without CFM (R2)**. We copy the group-level vector in $\mathbb{R}^{C'}$ for $H \times W$ times to form a tensor in $\mathbb{R}^{C' \times H \times W}$
with the same spatial size as the single-image feature in $\mathbb{R}^{C \times H \times W}$, and concatenate them along the channel dimension.

**Background of SISMs (R2)**. Inter consistency in common backgrounds is also exploited by our ICNet to well leverage
SISMs. Otherwise, the results of our ICNet on $F_\beta$/$S_\alpha$/MAE drop from $0.874/0.862/0.048$ to $0.864/0.855/0.051$ on
*iCoseg*, from $0.860/0.855/0.058$ to $0.849/0.847/0.064$ on *CoSal2015*, respectively.

**Why removing "Tree" from *MSRC* (R2)**? The SISMs of the "Tree" category (in *MSRC*) produced by previous SOD
methods do not contain any saliency information. This will make our ICNet fail since it is heavily based on SISMs.
To show the power of our ICNet with reasonable SISMs on Co-SOD, we remove this "Tree" category. Thanks to the
reviewers for this comment, which reminds us to take the failure of our ICNet on the "Tree" category as our major
limitation. Hence, we have discussed this problem as a new section of "Failure Case" in the revision of our submission.

**Using SISMs provided by old SOD methods (R3)**. Thanks for this suggestion. We trained our ICNet with the SISMs
produced by Global Contrast (GC) (Global Contrast based Salient Region Detection. In *CVPR*, 2011). As shown in
Table 1, our "ICNet with GC" still outperforms other methods, validating the effectiveness of our ICNet.

**Why RSCF is useful (R3)**? We observed via experiments that our ICNet with combined CSA map $A_k$ and normalized
feature map $F_k$ fails to distinguish pixels with similar but different categories. This is mainly due to the inconsistency
on the category dependence between $A_k$ and $F_k$: $A_k$ is category-independent and just reflects the initial co-saliency
scores, while $F_k$ is category-related and each pixel in it is a vector representing a specific category. Specifically, in
our ICNet the predictions mainly depend on category-independent $A_k$, indicating that the category information in $F_k$
is neglected. To tackle this inconsistency, we calculated the similarity between pairs of pixels in $F_k$ by utilizing the
category information in it, and transferred $F_k$ into category-independent SCF. We then rearranged the channel order of
SCF to obtain the RSCF, according to the co-saliency scores in $A_k$. These operations enable our ICNet well exploit
meaningful information from the rearranged self-correlation maps in SCF. In **Lines 269-278** of the main paper, the
ablation studies have validated the effectiveness of our RSCF. We have added this point to our revised submission.

**Why integrating SISMs performs better (R4)**? Taking SISMs as training targets forces Co-SOD models overfit to
inaccurate SISMs with performance drop, though pooling/normalization layers are used to dilute the inaccuracy. By
directly integrating SISMs into the encoder of the model, our ICNet avoids the problem of overfitting to inaccurate
SISMs, while the inherent inaccuracy in SISMs is largely alleviated by NMAP. The effectiveness of directly integrating
SISMs into our ICNet has been validated in the ablation study (**Lines 262-265**) of the main paper.

[Meta-Review · NeurIPS 2020]

The initial ratings were 5666. The main concerns were: 1) incremental novelty; 2) missing ablation on other datasets; 3) whether "Tree" group in MSRC experiments was also removed for other methods; 4) only small gains on some datasets In the response, authors clarified the contribution. They also provide additional ablation studies and provide reasons for the small gains on MSRC and iCoseg since they usually only contain one category of salient objects, whereas CoSal2015 has much larger proportion of multi-category saliency objects and the proposed method performs significantly better. After the response and discussion phase, all reviewers kept their original ratings. In particular, R2 was satisfied with most responses, except for the question about the "Tree" group being removed, which was not answered directly. The AC emailed the authors to clarify the point and they responded "Yes, for fair comparisons, in this paper, all methods (ours and the others) are evaluated on the MSRC without the "Tree" group.". Thus, the AC thinks this concern is addressed. Since all concerns were addressed, the AC recommends accept. Authors should update the paper according to the reviews and responses.